# Nociplastic Pain in Multiple Sclerosis Spasticity: Dermatomal Evaluation, Treatment with Intradermal Saline Injection and Outcomes Assessed by 3D Gait Analysis: Review and a Case Report

**DOI:** 10.3390/ijerph19137872

**Published:** 2022-06-27

**Authors:** Paolo De Blasiis, Giampaolo de Sena, Elisabetta Signoriello, Felice Sirico, Marta Imamura, Giacomo Lus

**Affiliations:** 1Section of Human Anatomy, Department of Mental and Physical Health and Preventive Medicine, University of Campania “Luigi Vanvitelli”, 80138 Naples, Italy; 2Villa Germana Clinica Ruesch, 80122 Naples, Italy; giampaolo.desena@gmail.com; 3Second Division of Neurology, Department of Advanced Medical and Surgical Sciences, University of Campania “Luigi Vanvitelli”, 80131 Naples, Italy; elisabetta.signoriello@unicampania.it (E.S.); giacomo.lus@unicampania.it (G.L.); 4Department of Public Health, University of Naples “Federico II”, 80131 Naples, Italy; sirico.felice@gmail.com; 5Instituto de Medicina Física e Reabilitação, Hospital das Clinicas HCFMUSP, Faculdade de Medicina, Universidade de São Paulo, São Paulo 05403-000, SP, Brazil; marta.imamura@fm.usp.br

**Keywords:** nociplastic pain, spasticity, multiple sclerosis, infiltrative treatment, gait analysis

## Abstract

Nociplastic pain has been introduced by the IASP as a third category of pain, distinct from nociceptive and neuropathic pain. Pathogenetically, it is considered to be a continuum of these two types of pain after becoming chronic. Repetitive peripheral painful stimulation causes a central sensitization with hypersensitivity of the corresponding spinal metamer or brain region. Therefore, signs of altered nociception, such as allodynia, may be found on the tissues of the related dermatome, myotome and sclerotome, and characterize nociplastic pain. This kind of pain was found in over 20% of people with multiple sclerosis (pwMS), a demyelinating autoimmune disease that affects the central nervous system. Nociplastic pain may be an amplifier of spasticity, the main pyramidal symptom that affects about 80% of pwMS. This article details the case of a 36-year-old woman with multiple sclerosis who was affected by spasticity and non-specific pain of the lower limbs, disabling on walking. Previous analgesic and muscle relaxant treatment had no benefits. The diagnosis of nociplastic pain on the cutaneous tissue of the anterolateral region of the left thigh and its treatment with intradermal normal saline injection on the painful skin area showed immediate and lasting effects on pain and spasticity, improving significantly the patient’s balance and walking, as assessed by a 3D motion analysis and rating scales.

## 1. Introduction

Nociplastic pain has recently been defined by the International Association for the Study of Pain (IASP) as the “Pain that arises from altered nociception despite no clear evidence of actual or threatened tissue damage causing the activation of peripheral nociceptors or evidence for disease or lesion of the somatosensory system causing the pain” [1]. It represents a third category of pain that is distinct from nociceptive pain, which is caused by ongoing inflammation and tissue damage, and neuropathic pain, which is caused by nerve damage. The pathogenetic mechanisms of nociplastic pain are not entirely understood, but there is evidence that peripheral painful chronic stimulation causes a central sensitization, with hypersensitivity of the corresponding spinal metamer [2,3]. As a consequence, dysesthesia, hyperalgesia and tactile or temperature allodynia may be found on a limited area of the body or on the tissue of the related dermatome, myotome and sclerotome; therefore, nociplastic pain may be considered as an overlap and “continuum” of the other two types of pain, more than a unique category of pain [2,3]. The clinical features of nociplastic pain may fluctuate both in location and intensity and be reversible or aggravated by physical activity, environmental stimuli and psycho-behavioral factors; moreover, central nervous system symptoms, such as fatigue, sleep and mood disturbances, cognitive impairment and hypersensitivity to external stimuli may be associated [2,3]. Nociplastic pain may be found in over 20% of people with multiple sclerosis (pwMS), a demyelinating autoimmune disease that affects the central nervous system [4]; about 80% of pwMS sufferers are affected by several degrees of spasticity [5], the most important symptom of the pyramidal disfunction, characterized by an involuntary muscle hyperactivity in the presence of central paresis [6]. Spasticity is considered to be the most disabling symptom when walking and the main cause of falls [7]; all types of pain may be amplifiers of spasticity [8]. Unfortunately, nociplastic pain is not often evaluated and treated, probably due to the recent definition of its clinical features and evaluation methods. A few studies proposed diagnostic procedures and tools that are useful for correct assessment [8], while others suggested different treatments for dermatomal hyperalgesia, such as acupuncture and manipulative medicine [9], mesotherapy [10] and intradermal saline infiltration [11], with an efficacy that is not clearly supported by strong evidence. In this article, we reported the evaluation method and a dermatomal infiltrative treatment for cutaneous nociplastic pain, showing the results by 3D gait analysis.

## 2. Case Report

A 36-year-old woman who was affected by multiple sclerosis (MS) and had a known motor condition of mild spastic paraparesis (left > right) came to our MS Center for a referred increased stiffness and pain of the left lower limb that interfered on ambulation and other motor daily life activities, forcing her to often use a walking aid. Previous analgesic and muscle relaxant treatment had no benefits (before systemic: Baclofene 10 mg, 1 per/die for 1 week, with increase in weakness without benefits and suspended; after focal: botulinum toxin on femural rectus and gastrocnemius left lower limb with transient benefits). MRI didn’t show new spinal cord or brain alterations with respect to the previous one. The patient was assessed by impairment-disability scales and a 3D-Gait Analysis at T0 (before treatment); T1 (5 min after treatment); T2 (after 5 days); and T3 (after 60 days). The scales that were used for the assessment included: the Expanded Disability Status Scale (EDSS) for disability degree; Medical Research Council (MRC) for strength; Modified Tardieu scale (MTS) for spasticity; goniometer for the ROM; Numeric Rating Scale (NRS) for pain intensity; pelvimeter for larger diameters of painful skin area; Berg Balance Scale (BBS) for balance; 10 Meter walking test (10 MWT) for gait speed; 2 Minute Walking Test for endurance; and 3D-gait analysis using a Helen Hayes MM marker set [12] to evaluate the spatial-temporal and kinematic parameters, such as Gait Profile Score (GPS) and nine Gait Variable Scores (GVS), both of which are frequently used to quantify deviation of the dynamic ROM from physiological ones during gait in multiple sclerosis [13,14]. After the first clinical and instrumental assessment (T0), the patient was treated with intradermal saline injections without undressing. the reflective skin markers (Figure 1b) were useful for the subsequent gait analysis (T1), in order to avoid repositioning bias. 

At T0, the patient showed weakness and severe spasticity of the left lower limb that was associated to a limitation of the knee and ankle range of movement (ROM), as well as cutaneous nociplastic pain on the antero-lateral region of the thigh, diagnosed with elicitation of the allodynia by a paperclip on the cutaneous tissues (Figure 1a; Table 1). After, the painful skin area was treated with normal saline injections (Figure 1b), 7.5 mL in 25 sites (0.3 for each site). The treatment showed immediate and lasting beneficial effects on pain and, consequently, on spasticity. Regarding the clinical scores, an extraordinary decrease in the pain, skin-painful area and spasticity; an increase in the active and passive ROM of the knee and ankle; the strength of the knee-flexor and dorsi-flexor muscles, balance, gait speed, and endurance were found after 5 min from treatment and subsequent evaluations at 5 and 60 days (Table 1). 

Regarding the instrumental analysis, time-distance parameters showed an increase in cycle duration, gait speed and stride length; moreover, a greater symmetry between the two lower limbs regarding the stance, swing phase and double support was found after treatment (Table 2). Regarding the kinematic values, a lower GPS and GVS, above all of the whole pelvic ROM, knee flexion and ankle dorsiflexion, were found bilaterally (Table 2). Eventually, the kinematic curves showed an increase in knee flexion and ankle dorsiflexion and a decrease in hip adduction and pelvic obliquity during the swing phase on the left lower limb (Figure 2). 

## 3. Discussion

The evaluation and treatment of cutaneous nociplastic pain in pwMS should be an important target of clinical practice because of the high prevalence of the disease and the percentage of patients that are affected by spasticity and this type of pain. In fact, the disability deriving from multiple sclerosis and spasticity is amplified by nociplastic pain. The present case report highlights how the patient had difficulty on balance, standing up from the chair and walking. The results of our study showed the significant improvement of both the impairment and disability scale, and of the kinematic parameters (Table 1 and Table 2; Figure 2), underlining not only a decrease in pain (NRS) and spasticity (MTS) and an increase in dorsiflexor and knee extensor muscles strength (MRC), but also the immediate and lasting beneficial effects on balance and walking, clinically (BBS, 10 MWT, 2 MWT) and instrumentally (GPS, GVS and kinematic curves) quantified. Therefore, these results demonstrate how the treatment of the skin-painful area may solve the trigger of spasticity amplification of the left lower limb.

Skin pain was first described in 1863 by Sir John Hilton [15], who defined an anatomical and neurophysiological law which said that “the nerve supplying a joint also supplies both the muscles that move the joint and the skin covering the articular insertion of those muscles”. Hilton observed that, in the case of arthritis, the nervous system was forced to rest those muscles and caused pain on the corresponding cutaneous area. This clinical observation was confirmed and deepened by many studies about dermatomal hyperalgesia in patients with knee osteoarthritis [16], myofascial pain [17] and trigger points [18], such as in Maigne Syndrome [19], in which all symptoms refer to central sensitization and nociplastic pain [3]. In clinical practice, many treatments are often used without full awareness for cutaneous pain, such as acupuncture and manipulative medicine [9,20], mesotherapy [10] and intradermal saline infiltration [11], with an efficacy that is not clearly supported by strong evidence. We hypothesized that any kind of mechanical (needling, manipulation) or biochemical (infiltration) stimulation of cutaneous receptors belonging to a spinal metamer that is hypersensitized by chronic pain, may have beneficial effects on nociplastic pain, resetting spinal neuronal circuits. In particular, we chose an intradermal saline injection because of the absence of side effects and, mainly, for the effective use reported in a previous and very interesting study [11] about the treatment of chronic pain in patients that were affected by herpes zoster. In this article, patients were treated with an intradermal saline infiltration about 2 cm lateral to the mid-spine, in relation to the nerve roots of a hypersensitized dermatome. The injection produced an initial augmented burst of pain that was followed immediately by profound, long-lasting pain relief. This old article promoted the use of saline injection in selected patients with chronic pain in our clinical practice; beneficial effects encouraged this therapeutic approach. Obviously, this is only a first case report about the treatment of nociplastic pain in a patient that is affected by multiple sclerosis and spasticity, even if carefully assessed by clinical scales and instrumental analysis. Further studies are necessary to better understand the mechanisms of onset and healing of this type of pain, and to validate evaluation methods and effective therapies.

## 4. Conclusions

The present study is the first to report a case of nociplastic pain in a patient with multiple sclerosis and spasticity, evaluated clinically and instrumentally before and after treatment with an intradermal saline injection. Our method, based on dermatomal evaluation and infiltrative therapy might be an effective and low-cost diagnostic-therapeutic proposal for the management of cutaneous nociplastic pain in patients with multiple sclerosis and spasticity.

## Figures and Tables

**Figure 1 ijerph-19-07872-f001:**
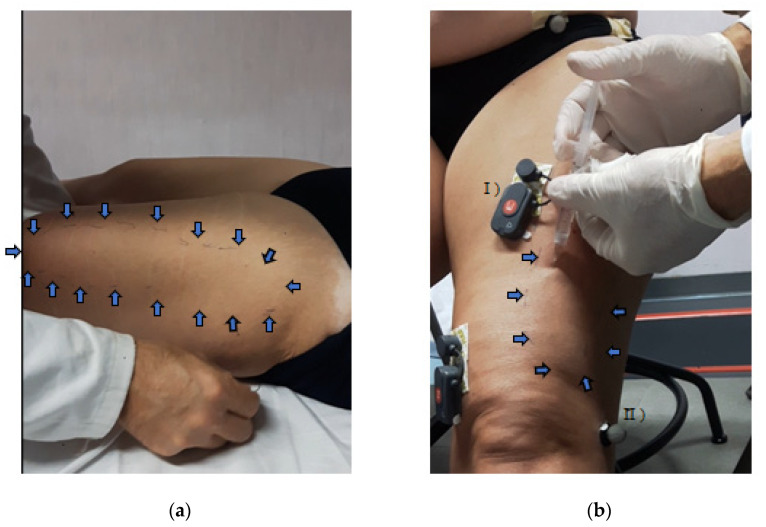
Evaluation (**a**) and treatment (**b**) of cutaneous area affected by nociplastic pain. (**a**) A paperclip was used to outline the painful skin area, marking with the black pen the area of transition from normal sensitivity to allodynia (the arrows highlight the black dotted lines that circumscribe the internal skin area affected by nociplastic pain). (**b**) Intradermal saline injections were performed for the treatment of the painful cutaneous area. (**b**-I) Surface-EMG; (**b**-II) reflective skin markers (medical devices used for 3D-gait analysis). Surface-EMG signal analysis are shown in the Appendix A.

**Figure 2 ijerph-19-07872-f002:**
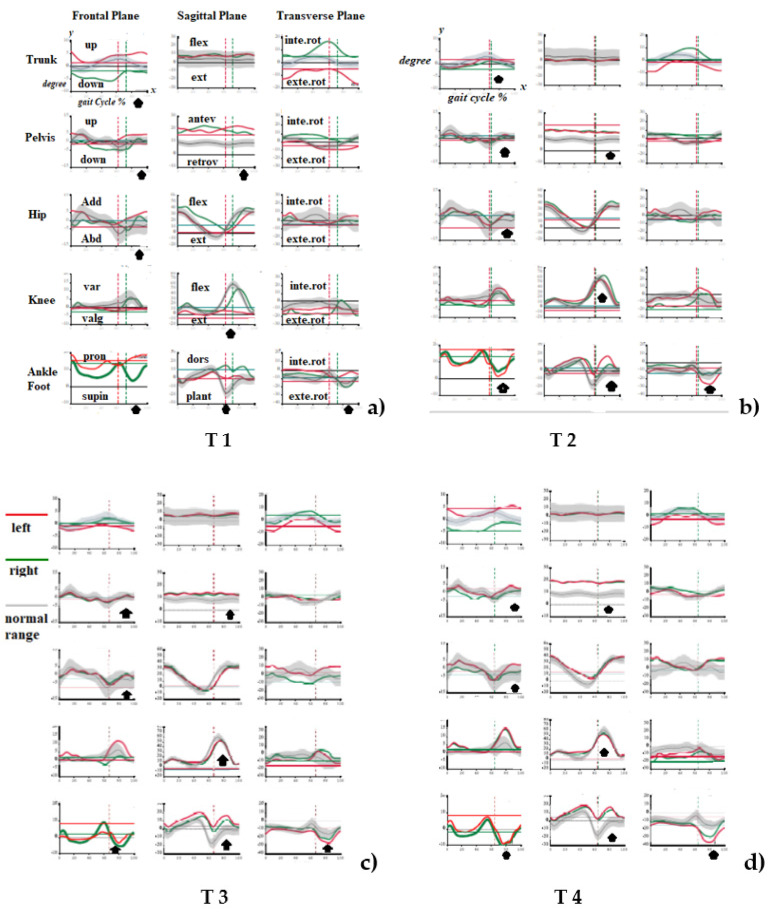
Trend of kinematic curves in three planes of motion before treatment at T0 (**a**), 5 min after treatment at T1 (**b**), after 5 days at T2 (**c**) and after 60 days at T3 (**d**). Each kinematic graph represents on the X-axis the percentage of gait cycle and on the Y-axis the motion degrees of each joint/body district (**b**). Figure note: black arrows indicate kinematic deviations from normal ranges at T0, and the improvement after treatment (at T1, T2, T3). Add (adduction); Abd (abduction); Var (varization); Valg (valgization); Pron (pronation); Supin (supination); flex (flexion); ext (extension); antev (anteversion); retrov (retroversion); dors (dorsiflexion); plant (plantiflexion); inte.rot (internal rotation); exte.rot (external rotation).

**Table 1 ijerph-19-07872-t001:** Clinical score before (at T0) and after treatment (at T1 after 5 min, at T2 after 5 days and at T3 after 60 days). Numbers in bold highlight a larger variation of the results.

Impairment Scales about Left Lower Limb	Timing of Assessment			
	T0	T1	T2	T3
** *MRC hip flexor* **	3	3	3	3
** *MRC hip extensor* **	4	4	4	4
** *MRC hip adductor* **	3	3	3	3
** *MRC hip abductor* **	3	3	3	3
** *MRC knee extensor* **	5	5	5	5
** *MRC knee flexor* **	2	**3**	**3**	**3**
** *MRC plantar flexor* **	3	3	3	3
** *MRC dorsiflexor* **	2	**3**	**3**	**3**
** *MTS knee extensor* **	5	**1**	**1**	**1**
** *MTS plantar flexor* **	4	**2**	**2**	**2**
***Active ROM of ankle with knee extended* (°)**	−18	**0**	**0**	**0**
***Passive ROM of ankle with knee extended* (°)**	−10	**10**	**10**	**10**
***Active ROM of knee* (°)**	0	**90**	**140**	**140**
***Passive ROM of knee* (°)**	90	**130**	**140**	**140**
**NRS *painful skin area***	10	**6**	**2**	**2**
***Size of the painful skin area* (cm^2^)**	20 × 7 = 140	**5 × 3 = 15**	**0**	**0**
**Disability scales**				
** *EDSS* **	6	**5.5**	**5**	**5**
** *BBS* **	38	**49**	**51**	**51**
***10 MWT* (sec)**	14.5	**10.8**	**10.2**	**11**
***2 MWT* (mt)**	96	**134**	**148**	**136**

**Table note:** MRC (Medical Research Council); MTS (Modified Tardieu Scale); ROM (Range of Movement); NRS (Numeric Rating Scale); EDSS (Expanded Disability Status Scale); BBS (Berg Balance Scale); 10 MWT (10 Meter Walking Test); 2 MWT (2 Minutes walking Test).

**Table 2 ijerph-19-07872-t002:** Time-distance and kinematic parameters of both lower limbs before treatment (at T0) and after treatment (at T1 after 5 min, at T2 after 5 days, at T3 after 60 days). Numbers in bold highlight a larger variation of the results.

t-d and k Parameters(T0—T1—T2)	T0LLL	T0RLL	T1LLL	T1RLL	T2LLL	T2RLL	T3LLL	T3RLL
	(mean ± SD)	(mean ± SD)	(mean ± SD)	(mean ± SD)	(mean ± SD)	(mean ± SD)	(mean ± SD)	(mean ± SD)
**Cadence** (step/min)	113.4 ±7.7	**101.55 ± 3.4**	**99.75 ± 3.07**	**103 ± 1.85**
**Cycle duration (s)**	1.05 ± 0.08	1.08 ± 0.08	**1.17 ± 0.04**	1.20 ± 0.05	**1.20 ± 0.03**	**1.21± 0.03**	**1.18 ± 0**	**1.15 ± 0.04**
**Gait speed (m/sec)**	0.803 ± 0.07	**0.875 ± 0.08**	**0.841** ± **0.05**	**0.833 ±** 0.02
**Stance phase (% g.c.)**	62.29 ± 5.48	72.28 ± 1.55	**66.82 ± 0.51**	**68.59 ± 1.38**	**66.51 ± 1.34**	**66.64 ±1.66**		
**Swing phase** (% g.c.)	37.71 ± 5.48	27.72 ± 1.55	**33.18 ± 0.51**	**31.41 ± 1.38**	**33.49 ± 1.34**	**33.36 ± 1.66**	**35.88 ± 1.6**	**35.2 ± 3.11**
**Double support (% g.c.)**	13.96 ± 2.48	21.7 ± 3.5	**16.15 ± 0.79**	**18.42 ± 1.28**	**15.97 ± 1.81**	**16.36 ± 1.4**	**14.12 ± 2.12**	**16.27 ± 2.64**
**Stride lenght (m)**	0.84 ± 0.13	0.87 ± 0.03	**1.03 ± 0.08**	**1.04 ± 0.08**	**0.99 ± 0.06**	**1.01 ± 0.04**	**0.98 ± 0.04**	**0.98 ± 0.04**
**Step width (m)**	0.11 ± 0.01	0.11 ± 0.01	0.11 ± 0.01	0.11 ± 0.01
**GPS** (°)	10.9 ± 0.2	9.4 ± 0.5	**6.3 ± 0.7**	**6.7 ± 0.6**	**6.8 ± 0.2**	**5.9 ± 0.4**	**8.3 ± 0.6**	**7.5 ± 0.2**
**GVS pelvic tilt** (°)	11.4 ± 0.6	10.8 ± 1.5	**6.5 ± 0.8**	**6.9 ± 1**	**4.3 ± 0.9**	**4.1 ± 0.7**	**9.7 ± 0.4**	**9.5 ± 0.3**
**GVS Pelvic rotation (°)**	6.2 ± 1.6	8.2 ± 1.7	**4.5 ± 1**	**5.6 ± 1.2**	**4.4 ± 1**	**3.9 ± 0.6**	**4.1 ± 0.6**	**4.4 ± 0.6**
**GVS Pelvic obliquity (°)**	2.9 ± 0.6	3.8 ± 0.4	**1.3 ± 0.3**	**2 ± 0.4**	**1.2 ± 0.1**	**1.6 ± 0.3**	**1.8 ± 0.2**	**2.8 ± 0.3**
**GVS Hip flex-extension** (°)	8.4 ± 1.7	13.4 ± 1.3	**7.0 ± 1.1**	**8.5 ± 0.7**	**5.9 ± 1**	**5.7 ± 0.7**	**7.5 ± 0.9**	**7.7 ± 0.8**
**GVS Hip abd-adduction** (°)	4.7 ± 0.4	3.7 ± 0.6	**2.1 ± 0.6**	**3 ± 0.4**	**1.8 ± 0.4**	**2.2 ± 0.3**	**2.6 ± 0.3**	**2.4 ± 0.3**
**GVS Hip rotation** (°)	7.1 ± 0.7	6.2 ± 0.4	7.1 ± 0.2	7 ± 0.5	8.5 ± 0.2	9.1 ± 0.2	10.1 **±** 0.2	4.4 **±** 0.6
**GVS Knee flex-extension** (°)	24.3 ± 1.4	12.7 ± 2.8	**6.8 ± 0.9**	**10.1 ± 1**	**6 ± 1**	**7 ± 1.6**	**7.1 ± 0.3**	**6.5 ± 0.9**
**GVS Ankle dorsiflexion (°)**	7.5 ± 1	13.3 ± 1.2	**6.1 ± 1.4**	**7.9 ± 1**	**7 ± 0.4**	**9.2 ± 0.4**	10.7 ± 1	11.1 ± 0.3
**GVS Foot progression (°)**	8.8 ± 2.3	4.4 ± 0.6	**5.9 ± 1.9**	4.4 ± 1.3	6.8 ± 0.8	4.1 ± 0.8	11.7 ± 2	8.4 ± 1.3

**Table note:** t-d (time-distance); k (kinematic); LLL (left lower limb); RLL (right lower limb); SD (standard deviation); GPS (Gait Profile Score); GVS (Gait Variable Score).

## Data Availability

The data that support the findings of this study are available from the corresponding author upon reasonable request.

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
