# Peer review of "Nociplastic Pain in Multiple Sclerosis Spasticity: Dermatomal Evaluation, Treatment with Intradermal Saline Injection and Outcomes Assessed by 3D Gait Analysis: Review and a Case Report"

_ijerph, 2022, doi:10.3390/ijerph19137872_

Round 1

Reviewer 1 Report

Thank you for the opportunity to review your manuscript.  Nociplastic pain is a curious type of pain and I appreciate this case study as we begin to look more closely at the etiology of this third pain type.  I am not sure if the health care world accepts this pain as true pain but as a sufferer of pain that cannot be described by neuropathic or nociceptive I recognize the need for further research and publications.

The following are suggestions for the authors to take into consideration:

Line 26 36-year-old woman

Line 67 36-year-old woman

Line 88 Figure 1 It is advantageous for figures to have sufficient information provided so that a reader does not need to read the entire article but can simply read the tables and figures and summarize the important findings of the manuscript.  I suggest placing additional information such as what evaluations were done with the patient posed in (a).  Also is the nociplastic pain linear?  Meaning it is ONLY along the dotted line with black pen?  It would be helpful to demarcate this pen lines with black arrows so that it is easier for the reader to visualise. In (b) please provide an through explanation of the equipment pictured on the patient and the injection being given.  Please write a statement about the nociplastic pain location and the pen marks.

Line 93 please spell out “table” so it is consistent as in line 99

Line 122-123 This is figure 2, not figure 1.  Please consider selecting a few representative curves and using text to describe the results of the others.  The format which is used is not easy to read or see or analyze.  If representation is used then the reader can access additional curves in the supplemental material.  Figure 2 is not useful as submitted and I fear readers will not look at these results.  It is too busy, too small, too hard to read.  Please also be careful of red type as it is very challenging for color-impaired individuals to see.  I would offer black arrows for indicating kinematic deviations.

Line 131 Multiple Sclerosis not Sclerosis Multiple

Discussion may wish to include the authors’ proposed theory as to why they saw the differences they recorded.  Why a saline injection?  What is the proposed mechanism of action of the fluid?

Lines 143-144 Please consider re-phrasing this sentence

The following are suggestions for the authors to take into consideration:

Line 26 36-year-old woman

Line 67 36-year-old woman

Line 88 Figure 1 It is advantageous for figures to have sufficient information provided so that a reader does not need to read the entire article but can simply read the tables and figures and summarize the important findings of the manuscript.  I suggest placing additional information such as what evaluations were done with the patient posed in (a).  Also is the nociplastic pain linear?  Meaning it is ONLY along the dotted line with black pen?  It would be helpful to demarcate this pen lines with black arrows so that it is easier for the reader to visualise. In (b) please explain the expquiment pictured and the injection being given.

Author Response

 Response to Reviewer 1

Thank you for the opportunity to review your manuscript. Nociplastic pain is a curious type of pain and I appreciate this case study as we begin to look more closely at the etiology of this third pain type.  I am not sure if the health care world accepts this pain as true pain but as a sufferer of pain that cannot be described by neuropathic or nociceptive. I recognize the need for further research and publications.

The authors thank the reviewer for his nice comment and interest about this study.

The following are suggestions for the authors to take into consideration:

Line 26 36-year-old woman

The text has been revised accordingly.

Line 67 36-year-old woman

Ok this has been done.

Line 88 Figure 1 It is advantageous for figures to have sufficient information provided so that a reader does not need to read the entire article but can simply read the tables and figures and summarize the important findings of the manuscript.  I suggest placing additional information such as what evaluations were done with the patient posed in (a).  Also is the nociplastic pain linear?  Meaning it is ONLY along the dotted line with black pen?  It would be helpful to demarcate this pen lines with black arrows so that it is easier for the reader to visualise. In (b) please provide an through explanation of the equipment pictured on the patient and the injection being given.  Please write a statement about the nociplastic pain location and the pen marks.

The authors would like to thank the reviewer for his useful and thoughtful suggestions. The explanation of the image has been thorough and additional informations has been integrated. Moreover, the picture has been corrected according to the suggestions.

Line 93 please spell out “table” so it is consistent as in line 99

 Ok, this has now been fixed.

Line 122-123 This is figure 2, not figure 1.  Please consider selecting a few representative curves and using text to describe the results of the others.  The format which is used is not easy to read or see or analyze.  If representation is used then the reader can access additional curves in the supplemental material.  Figure 2 is not useful as submitted and I fear readers will not look at these results.  It is too busy, too small, too hard to read.  Please also be careful of red type as it is very challenging for color-impaired individuals to see.  I would offer black arrows for indicating kinematic deviations.

The authors thank the reviewer for the suggestion and respect his point of view and impressions on the figure 2. Nevertheless, according to us, it’s necessary to show all kinematic graphs because the changes before and after treatment occur in all three planes of motion (frontal, sagittal and transverse). We tried to improve the figure, inserting more explanation on the graphs and under the figure. Moreover we added all kinematic reports and electromyographic graphs of T0 and T1 in the supplemental material.

Line 131 Multiple Sclerosis not Sclerosis Multiple

Ok, this has now been fixed.

Discussion may wish to include the authors’ proposed theory as to why they saw the differences they recorded.  Why a saline injection?  What is the proposed mechanism of action of the fluid?

The choice of saline injection derived mainly from the study cited in the text*, where the author used it for the treatment of chronic skin pain in patients with herpes zoster. Moreover, the absence of side effects and subsequent beneficial effects in treatment of non-specific chronic pain by various homotoxicological and homeopathic active ingredients (all dissolved in normal saline) suggested us the treatment of intradermal saline injection.

We hypothesized that any kind of mechanical (needling, manipulation) or biochemical (infiltration) stimulation of cutaneous receptors, belonging to spinal metamer hypersensi-tized by chronic pain, may have beneficial effects on nociplastic pain, resetting spinal neuronal circuits.

* Collins HW. Obtaining automated diagnostic information and pain relief. Med Hypotheses. 1995 Oct;45(4):389-91. doi: 10.1016/0306-9877(95)90101-9. PMID: 8577304

Reviewer 2 Report

The manuscript entitled “ Nociplastic pain in Multiple Sclerosis spasticity: dermatomal evaluation, treatment with intradermal saline infiltration and outcomes assessed by 3D gait analysis. Review and a case report” describes a case report on nociplastic pain in a Multiple Sclerosis and spasticity patient who evaluated before and after the treatment with intradermal saline infiltration. The method described in the manuscript could be useful to develop a diagnostic-therapy for the management of cutaneous nociplastic pain in patients with Multiple Sclerosis and spasticity. In my opinion, this manuscript is suitable to publish in journal “International Journal of Environmental Research and Public Health” in its current form.

Author Response

Response to Reviewer 2

The manuscript entitled “ Nociplastic pain in Multiple Sclerosis spasticity: dermatomal evaluation, treatment with intradermal saline infiltration and outcomes assessed by 3D gait analysis. Review and a case report” describes a case report on nociplastic pain in a Multiple Sclerosis and spasticity patient who evaluated before and after the treatment with intradermal saline infiltration. The method described in the manuscript could be useful to develop a diagnostic-therapy for the management of cutaneous nociplastic pain in patients with Multiple Sclerosis and spasticity. In my opinion, this manuscript is suitable to publish in journal “International Journal of Environmental Research and Public Health” in its current form.

The authors would like to thank the reviewer for his nice and thoughtful comments. We are very happy that our study was appreciated and considered satisfactory.

Reviewer 3 Report

The manuscript reported treatment with intradermal saline infiltration for cutaneous nociplastic pain  in Multiple Sclerosis spasticity

I have no general objection to the manuscript. The manuscript was written according to the general reporting Case report for the note's description. Similarly, research procedures are very detailed and insightful.

Nevertheless, I believe that despite the unequivocal results of the study, the conclusions should be much more cautious. Especially due to the (emphasized by the authors) lack of scientific sources with which to confront the obtained results.

The authors rightly stated that further research is needed to confirm the obtained results.

I believe that the discussion lacks an attempt to explain the mechanism of action of the applied therapy and the polemics with the possible placebo effect, what needs to be added.

I also have some technical notes:

Please read the text carefully and correct any punctuation errors, e.g.:

- in line 76 there is no space between "60" and "days", similarly in lines 77 and 79.

- line 93 is missing the dot after the "fig"

- line 113 missing spaces after "fig"

line 115

The table should be formatted, e.g., it should be (T0 - T1-T2) instead of (T0 - T1 - T2).

Correct the first column so that there is no unit transfer or move all units to the next row.

Author Response

Response to Reviewer 3

The manuscript reported treatment with intradermal saline infiltration for cutaneous nociplastic pain  in Multiple Sclerosis spasticity

I have no general objection to the manuscript. The manuscript was written according to the general reporting Case report for the note's description. Similarly, research procedures are very detailed and insightful.

Nevertheless, I believe that despite the unequivocal results of the study, the conclusions should be much more cautious. Especially due to the (emphasized by the authors) lack of scientific sources with which to confront the obtained results.

The authors rightly stated that further research is needed to confirm the obtained results.

I believe that the discussion lacks an attempt to explain the mechanism of action of the applied therapy and the polemics with the possible placebo effect, what needs to be added.

The authors would like to thank the reviewer for his nice comments and useful suggestions. We are in agreement with reviewer about the possible placebo effect, but the previous treatments with botulinum toxin injection and with muscle-relaxant (Baclofene) showing minimal and transient beneficial effects, suggested us that the long lasting pain relief comes from treatment of painful skin area with intradermal saline injection.

About the mechanism of action, we hypothesized that mechanical (injection) and biochemical (normal saline) stimulation of cutaneous receptors, belonging to spinal metamer hypersensi-tized by chronic pain, may have beneficial effects on nociplastic pain, resetting spinal neuronal circuits.

I also have some technical notes:

Please read the text carefully and correct any punctuation errors, e.g.:

- in line 76 there is no space between "60" and "days", similarly in lines 77 and 79.

The text has been revised accordingly.

- line 93 is missing the dot after the "fig"

Ok this has been done.

- line 113 missing spaces after "fig"

Ok this has been done.

line 115

The table should be formatted, e.g., it should be (T0 - T1-T2) instead of (T0 - T1 - T2).

Correct the first column so that there is no unit transfer or move all units to the next row.

The text has been revised according to reviewers’ suggestions.
